# Overlap matrix completion for predicting drug-associated indications

**Mengyun Yang[1,2], Huimin Luo[1], Yaohang Li[3], Fang-Xiang Wu[4], Jianxin Wang[1]** *

**1** School of Computer Science and Engineering, Central South University, Changsha, Hunan, P.R. China, **2** Provincial Key Laboratory of Informational Service for Rural Area of Southwestern Hunan, Shaoyang University, Shaoyang, Hunan, P.R. China, **3** Department of Computer Science, Old Dominion University, Norfolk, Virginia, United States of America, **4** Division of Biomedical Engineering, University of Saskatchewan, Saskatoon, Saskatchewan, Canada

* jxwang@mail.csu.edu.cn

**Data Availability Statement:** All relevant data can be obtained from the website https://github.com/BioinformaticsCSU/OMC.

**Funding:** This research was supported by the National Natural Science Foundation of China

## Abstract

Identification of potential drug–associated indications is critical for either approved or novel drugs in drug repositioning. Current computational methods based on drug similarity and disease similarity have been developed to predict drug–disease associations. When more reliable drug- or disease-related information becomes available and is integrated, the prediction precision can be continuously improved. However, it is a challenging problem to effectively incorporate multiple types of prior information, representing different characteristics of drugs and diseases, to identify promising drug–disease associations. In this study, we propose an overlap matrix completion (OMC) for bilayer networks (OMC2) and tri-layer networks (OMC3) to predict potential drug-associated indications, respectively. OMC is able to efficiently exploit the underlying low-rank structures of the drug–disease association matrices. In OMC2, first of all, we construct one bilayer network from drug-side aspect and one from disease-side aspect, and then obtain their corresponding block adjacency matrices. We then propose the OMC2 algorithm to fill out the values of the missing entries in these two adjacency matrices, and predict the scores of unknown drug–disease pairs. Moreover, we further extend OMC2 to OMC3 to handle tri-layer networks. Computational experiments on various datasets indicate that our OMC methods can effectively predict the potential drug–disease associations. Compared with the other state-of-the-art approaches, our methods yield higher prediction accuracy in 10-fold cross-validation and *de novo* experiments. In addition, case studies also confirm the effectiveness of our methods in identifying promising indications for existing drugs in practical applications.

## Author summary

This work introduces a computational approach, namely overlap matrix completion (OMC), to predict potential associations between drugs and diseases. The novelty of OMC lies in constructing an efficient framework of incorporating multiple types of prior information in bilayer and tri-layer networks. OMC for bilayer networks (OMC2) can approximate the low-rank structures of the drug–disease association matrices from both

(Grant No. 61972423, 61802113, and 61420106009), the Graduate Research Innovation Project of Hunan (Grant No. CX20190125), the General Project of Hunan Education Department (Grant No. 17C1434), Hunan Provincial Science and technology Program (No. 2018wk4001), and 111 Project (No. B18059). The funders had no role in study design, data collection and analysis, decision to publish, or preparation of the manuscript.

**Competing interests:** The authors have declared that no competing interests exist.

drug-side and disease-side. In addition, we further improve the prediction accuracy by extending OMC to handle tri-layer networks and develop its corresponding algorithm (OMC3). To evaluate the performance of OMC2 and OMC3, we conduct 10-fold cross-validation and *de novo* experiments on three datasets. Our computational results demonstrate that both OMC2 and OMC3 generally outperform five state-of-the-art methods in terms of ROC curve, PR curve, and top-ranked predictions.

## Introduction

The development of new drugs is extremely time-consuming and expensive [1]. It is reported that the average time of developing a new drug is more than 13.5 years and the cost exceeds $1.8 billion dollars [2], while only a relatively small number of novel drugs are approved by US Food and Drug Administration (FDA) each year. Identifying new uses of existing drugs, known as drug repositioning, has been popularly used for the pharmaceutical industry and research community. Since the existing drugs have already owned safety, efficacy, and toleration data after numerous experiments and clinical trials, identifying new and reliable indications for commercialized drugs can sharply reduce time and costs. In addition, some successfully repositioned drugs, such as raloxifene, sildenafil, and thalidomide, have produced great revenues for their patent companies. Hence, drug repositioning is an important strategy of drug discovery in pharmaceutical industry.

The computational methods for drug repositioning have received much attention recently, as the traditional manual experimental investigation is complicated and inefficient. In recent years, many types of computational approaches have been proposed, including semantic inference, network-based analysis, and machine learning. The network-based methods are one of the popularly-used approaches to identify potential drug–disease associations. Based on the guilt-by-association principle, Wang *et al.* constructed a heterogenous graph between drug and target and proposed the HGBI (Heterogeneous Graph Based Inference) algorithm to predict potential drug–target interactions [3]. The HGBI algorithm is also used for prediction of drug–disease associations [4]. Based on the propagation flow algorithm, Martinez *et al.* proposed a network-based prioritization method named DrugNet for drug repositioning [5]. The DrugNet algorithm can perform both disease–drug and drug–disease prioritization by integrating drug, disease, and target information. In [6], the MBiRW method addressed the drug-repositioning problem by applying a bi-random walk algorithm on heterogeneous network with comprehensive similarity measures for drugs and diseases, obtained by utilizing logistic function [7] and ClusterONE [8].

Machine learning methods have attracted a lot of attention in recent years. Based on the common assumption that similar drugs tend to connect with similar diseases, Gottlieb *et al.* calculated five drug–drug similarity measures and two disease–disease similarity measures for drug-associated indication prediction, and presented a method (PREDICT) to identify potential drug indications for approved drugs [9]. Integrating chemical structure, drug–target interaction, and side-effect data, Wang *et al.* presented an approach called PreDR for drug–disease association prediction [10]. PreDR treated the prediction problem as a binary classification problem by defining a kernel function and applying an SVM-based learning algorithm. In [11], a matrix factorization model was developed to predict new indications for known drugs by incorporating the interaction network of genes. Luo *et al.* proposed a drug repositioning recommendation system (DRRS) [12]. Specifically, a heterogeneous network was constructed by integrating drug similarities, disease similarities, and drug–disease associations and the

adjacency matrix of the large-scale heterogeneous network was considered as a low-rank matrix. The singular value thresholding algorithm (SVT) [13] was implemented to complete the missing entries of a drug–disease association matrix. Yang *et al*. further proposed a bounded nuclear norm regularization (BNNR) model [14], not only tolerating the noisy similarities of drugs and diseases by employing regularization, but also ensuring that all predicted values are within the interval of [0, 1]. However, the computational cost of both DRRS and BNNR increases sharply when target (protein/gene) information is incorporated into the heterogeneous drug–disease network.

In this study, we propose an overlap matrix completion for bilayer networks (OMC2) and tri-layer networks (OMC3) to predict potential indications for approved and new drugs. We design two different networks from drug-side aspect and disease-side aspect, instead of constructing a large-scale heterogeneous drug–disease network. This can significantly reduce the computational complexity for matrix completion. Meanwhile, a BNNR model [14] developed in our previous work is implemented to fill out the missing entries in the block adjacency matrix of these networks. We evaluate the performance of OMC2 and OMC3 in three different datasets and compare them with five latest approaches for drug repositioning. Our computational results show that our methods yield better accuracy in predicting potential drug–disease associations.

## Materials and methods

In this section, we introduce OMC for bilayer networks (OMC2) and tri-layer networks (OMC3) to identify potential indications for both known and novel drugs. First of all, a concise description of experimental datasets is presented. In bilayer heterogeneous networks, we integrate drug–drug, disease–disease, and drug–disease information. In tri-layer heterogeneous networks, besides the above three kinds of data, drug–protein and disease–protein associations are considered. Then, we present the OMC2 algorithm for drug–disease bilayer networks to predict novel drug–disease associations. Finally, we extend OMC2 to an OMC3 algorithm in handling the tri-layer networks, where the target-related information is also incorporated.

### Datasets

To construct bilayer and tri-layer networks, we collected drug, disease, and target protein information from published literatures and related authoritative databases. The approaches to collect association information and to compute similarity are described below.

*Drug–disease associations*. Confirmed drug–disease associations were obtained from the supplementary material of [9], which was admittedly treated as the gold standard dataset. There were 1, 933 associations between 593 drugs registered from DrugBank [15] and 313 diseases listed in the Online Mendelian Inheritance in Man (OMIM) database [16].

*Drug–drug similarity*. Drug–drug similarities were calculated based on chemical structures. The Canonical Simplified Molecular Input Line-Entry System (SMILES) [17] of these 593 drugs were downloaded from DrugBank. Then, the Chemical Development Kit (CDK) [18] was utilized to compute hashed fingerprints for each drug with default parameters. Finally, the similarity between two drugs was measured by the Tanimoto score [19] in the range of [0, 1].

*Disease–disease similarity*. Disease–disease similarities were computed by MimMiner [20], which identifies similarity of appearance of MESH (medical subject headings vocabulary) terms between two diseases in medical descriptions from the OMIM database. In the MimMiner program, the disease–disease similarity was normalized to the interval of [0, 1].

*Drug–protein interactions.* The interactions between drugs and proteins were collected from DrugBank. We collected 3, 184 drug–target (protein) interactions between 576 relevant drugs of the gold standard dataset and 975 proteins.

*Disease–protein associations.* We collected disease–protein associations in two steps. In the first step, we downloaded the interactions between diseases included in the gold standard dataset and genes from CTD [21], and the total of 475 disease–gene interactions were collected. Secondly, these genes were mapped into 849 proteins in UniprotKB database [22]. There were 1, 066 associations between 166 diseases and 849 proteins at last.

## OMC algorithm for bilayer networks

**Two drug–disease bilayer networks and corresponding adjacency matrices.** We construct two heterogeneous drug–disease bilayer networks. One is composed of a drug–drug network and a drug–disease network and the other is of a disease–disease network and a drug–disease network. Fig 1 shows the workflow for constructing two bilayer networks and their corresponding block adjacency matrices.

For the drug–drug network with $m$ drug nodes, let $A_{RR} \in \mathbb{R}^{m \times m}$ be its adjacency matrix, where element $(A_{RR})_{ij}$ represents the similarity between drugs $r_i$ and $r_j$. Similarly, $A_{DD} \in \mathbb{R}^{n \times n}$ is the adjacency matrix of the disease–disease network with $n$ disease nodes, where $(A_{DD})_{ij}$

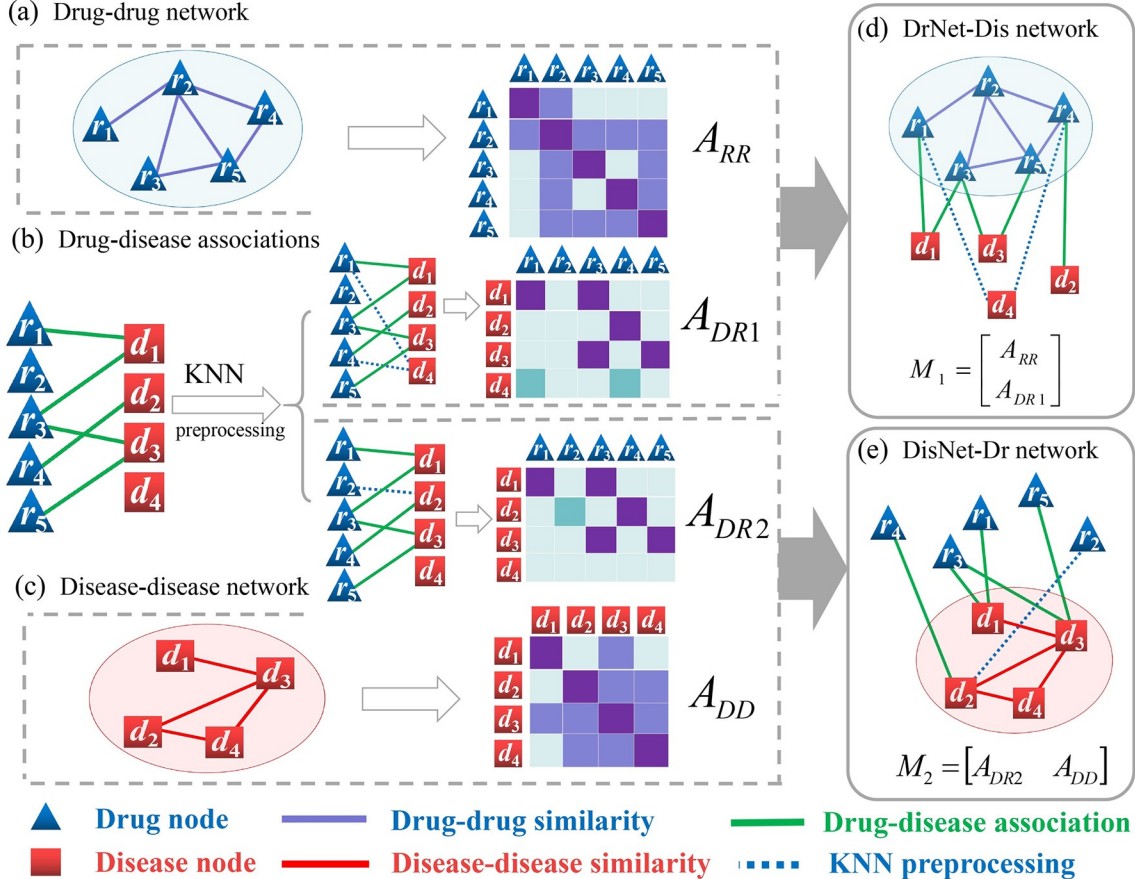

**Fig 1. The workflow of constructing the DrNet-Dis network and the DisNet-Dr network.** (a) Drug–drug network and its similarity matrix. (b) Drug–disease associations and KNN preprocessing. (c) Disease–disease network and its similarity matrix. (d) DrNet-Dis network and its block adjacency matrix. (e) DisNet-Dr network and its block adjacency matrix.

denotes the similarity between diseases $d_i$ and $d_j$. For the drug–disease network, let $A_{DR} \in \mathbb{R}^{n \times m}$ be its adjacency matrix (drug–disease association matrix), where $(A_{DR})_{ij}$ is set to 1 if there exists an experimentally validated association between $d_i$ and $r_j$, otherwise 0.

*DrNet-Dis network*. The DrNet-Dis network, illustrated in Fig 1(a), 1(b) and 1(d), is constructed by integrating the drug–drug network and the drug–disease network. For the sake of generality in applications, we take some novel disease nodes into account, which are not associated with any known drug node. For instance, $d_4$ is a new disease node in Fig 1(b), and the corresponding row of $A_{DR}$ is a zero vector, which causes difficulty in matrix completion and affects the performance of prediction. To address this cold-start problem, we conduct a *K-Nearest Neighbor (KNN) preprocessing* step for these new diseases. Specifically, for each novel disease $d_p$, $K$ nearest neighbor diseases of $d_p$ are picked based on their disease similarities in descending order. We update the corresponding row vector of disease $d_p$ in the drug–disease association matrix by filling out a part of weighted association information. The detail of the *KNN preprocessing* algorithm is described by Algorithm 1. After the *KNN preprocessing* step, an updated drug–disease association matrix $A_{DR1}$ is obtained and the block adjacency matrix $M_1 \in \mathbb{R}^{(m+n) \times m}$ of the DrNet-Dis network is presented as follows,

$$M_1 = \begin{bmatrix} A_{RR} \\ A_{DR1} \end{bmatrix}.$$

*DisNet-Dr network*. The DisNet-Dr network, demonstrated by Fig 1(b), 1(c) and 1(e), is constructed by integrating the disease–disease network and the drug–disease network. For some novel drugs (e.g., drug $r_2$ in Fig 1(b)), the corresponding columns of $A_{DR}$ are zero vectors. Similarly, the *KNN preprocessing* step is also implemented for these new drugs by Algorithm 1, and a new corresponding association matrix $A_{DR2}$ is developed. Finally, the block adjacency matrix $M_2 \in \mathbb{R}^{n \times (m+n)}$ of the DisNet-Dr network is denoted as follows,

$$M_2 = \begin{bmatrix} A_{DR2} & A_{DD} \end{bmatrix}.$$

Actually, the above *KNN preprocessing* step is not required if there is no novel disease or drug node. $M_1$ and $M_2$ are the to-be-complete matrices.

**Algorithm 1**: KNN Preprocessing Algorithm

**Input:** The drug similarity matrix $A_{RR} \in \mathbb{R}^{m \times m}$, the disease similarity matrix $A_{DD} \in \mathbb{R}^{n \times n}$, the disease-drug association matrix $A_{DR} \in \mathbb{R}^{n \times m}$ may contain some zero rows or columns, and the neighborhood size $K$.

**Output:** Updated $A_{DR1}$ and $A_{DR2}$.

1. Initialize $A_{DR1} = A_{DR}$ and $A_{DR2} = A_{DR}$;

2. Find index numbers of all zero rows of the matrix $A_{DR1}$, which are denoted as $\{i_1, i_2, \ldots, i_s\} \subset \{1, 2, \ldots, m\}$. $D_0 = \{d_{i_1}, d_{i_2}, \ldots, d_{i_s}\}$ represents the corresponding disease set. /* Entries of $D_0$ actually are novel diseases, where $d_{i_1}$ represents $i_1$-th disease in all diseases.*/

**for** each disease $d_p \in D_0$ **do**

3. $U = KNN(A_{DD}, K, d_p)$; /* *KNN* is a function for finding the $K$ nearest neighbors of disease node $d_p$ based on similarity matrix $A_{DD}$ in descending order.*/

4. $S_d = \sum\limits_{d_u \in U} A_{DD}(d_p, d_u)$;

5. $A_{DR1}(p, :) = \sum\limits_{d_u \in U} \frac{A_{DD}(d_p, d_u)}{S_d} * A_{DR}(d_u, :)$;

/*$A_{DR1}(p, :)$ notes the $p$-th row of matrix $A_{DR1}$ and the denominator is the normalization term.*/

**end for**

6. Find index numbers of all zero columns of the matrix $A_{DR2}$, which are denoted as $\{j_1, j_2, \ldots, j_t\} \subset \{1, 2, \ldots, n\}$. $R_0 = \{r_{j_1}, r_{j_2}, \ldots, r_{j_t}\}$ represents the corresponding drug set. /*Entries of $R_0$ actually are novel drugs, where $r_{j_1}$ represents the $j_1$-th drug in all drugs.*/
**for** each drug $r_q \in R_0$ **do**
7. $V = KNN(A_{RR}, K, r_q)$; /* KNN is a function for finding the $K$ nearest neighbors of drug node $r_q$ based on similarity matrix $A_{RR}$ in descending order.*/
8. $S_r = \sum\limits_{r_v \in V} A_{DD}(r_q, r_v)$;

9. $A_{DR2}(:, q) = \sum\limits_{r_v \in V} \frac{A_{RR}(r_q, r_v)}{S_r} * A_{DR}(:, r_v)$;

/*$A_{DR2}$(:, q) notes the $q$-th column of matrix $A_{DR2}$ and the denominator is the normalization term.*/
**end for**
10. **return** $A_{DR1}$ and $A_{DR2}$.

**BNNR model.** Matrix completion, whose goal is to recover the missing elements of matrix from only a few observations, has been widely used in many applications. Under the low-rank assumption, matrix completion is generally formulated as the following nuclear norm minimization problem

$$\min_X \; \|X\|_*$$
$$s.t. \; \mathcal{P}_\Omega(X) = \mathcal{P}_\Omega(M). \tag{1}$$

where $\|X\|_*$ denotes the nuclear norm of $X$, which is defined as the sum of all singular values of $X$. $M$ is the incomplete matrix, $\Omega$ is a set including index pairs $(i, j)$ of all known elements in $M$, and $\mathcal{P}_\Omega$ is the projection operator projecting matrix $X$ onto $\Omega$, which is defined as

$$(\mathcal{P}_\Omega(X))_{ij} = \begin{cases} X_{ij}, & (i, j) \in \Omega \\ 0. & (i, j) \notin \Omega \end{cases}$$

In the drug–disease association matrix, the entry value 1 denotes an experimentally validated indication while 0 indicates the association has not been validated yet. As a result, the predicted drug–disease association values are expected to fall in the interval of [0, 1], indicating the likelihood of being a true association. Therefore, a predicted value beyond the [0, 1] range is meaningless in the context of the application. To enforce the predicted values within the interval of [0, 1], a bounded constraint is added into the matrix completion model. In addition, due to the large amount of "noise" when calculating drug similarity and disease similarity, we relax the constraint satisfaction condition by incorporating a regularization term. As a result, we have proposed the bounded nuclear norm regularization (BNNR) described in [14] as follows,

$$\min_X \; \|X\|_* + \frac{\alpha}{2} \|\mathcal{P}_\Omega(X) - \mathcal{P}_\Omega(M)\|_F^2$$
$$s.t. \; 0 \le X \le 1. \tag{2}$$

where $\alpha > 0$ is a harmonic parameter to balance the nuclear norm and the error term and $0 \le X \le 1$ represents $0 \le X_{ij} \le 1$ for all $i, j$. A simple and effective algorithm is designed to solve model (2) by using the alternating direction method of multipliers (ADMM). By introducing a

new splitting matrix $W$, (2) can be formulated as the following equivalent form,

$$\min_{X} \ \|X\|_* + \frac{\alpha}{2}\|\mathcal{P}_\Omega(W) - \mathcal{P}_\Omega(M)\|_F^2$$

$$s.t. \ X = W,$$

$$0 \leq W \leq 1.$$

(3)

The augmented Lagrangian function of model (3) is

$$\mathcal{L}(W, X, Y, \alpha, \beta) = \|X\|_* + \frac{\alpha}{2}\|\mathcal{P}_\Omega(W) - \mathcal{P}_\Omega(M)\|_F^2$$

$$+ Tr(Y^T(X - W)) + \frac{\beta}{2}\|X - W\|_F^2,$$

(4)

where $Y$ is the Lagrange multiplier and $\beta > 0$ is the penalty parameter. By applying ADMM, we can obtain the following iterative scheme:

$$W_{k+1} = \arg\min_{0 \leq W \leq 1} \mathcal{L}(W, X_k, Y_k, \alpha, \beta),$$

(5)

$$X_{k+1} = \arg\min_{X} \mathcal{L}(W_{k+1}, X, Y_k, \alpha, \beta),$$

(6)

$$Y_{k+1} = Y_k + \beta(X_{k+1} - W_{k+1}).$$

(7)

We use the inverse operator [23] to solve Eq (5) and acquire a closed-form solution $W^*$ as follows,

$$W^* = (\mathcal{I} - \frac{\alpha}{\alpha + \beta}\mathcal{P}_\Omega)(\frac{1}{\beta}Y_k + \frac{\alpha}{\beta}\mathcal{P}_\Omega(M) + X_k),$$

where $\mathcal{I}$ denotes the identity operator. Moreover, to limit the element values of $W_{k+1}$ in the interval of $[0, 1]$, we utilize the following projection operator

$$W_{k+1} = \mathcal{Q}_{[0,1]}(W^*),$$

(8)

where $\mathcal{Q}_{[0,1]}$ is defined as

$$(\mathcal{Q}_{[0,1]}(W^*))_{ij} = \begin{cases} 1, & W_{ij}^* > 1 \\ W_{ij}^*, & 0 \leq W_{ij}^* \leq 1 \\ 0. & W_{ij}^* < 0 \end{cases}$$

By rearranging the terms of (6), we have

$$X_{k+1} = \arg\min_{X} \|X\|_* + \frac{\beta}{2}\left\|X - (W_{k+1} - \frac{1}{\beta}Y_k)\right\|_F^2$$

$$= \mathcal{D}_{\frac{1}{\beta}}(W_{k+1} - \frac{1}{\beta}Y_k),$$

(9)

where $\mathcal{D}_\tau(X)$ is the singular value shrinkage (SVT) operator [13] [24]. Specifically, SVT

operator is defined as

$$\mathcal{D}_{\tau}(X) = \sum_{i=1}^{\sigma_i \geq \tau}(\sigma_i - \tau)u_i v_i^T,$$

where $\sigma_i$ is the $i$th singular value of $X$ larger than threshold $\tau$, while $u_i$ and $v_i$ are the left and right singular vectors corresponding to $\sigma_i$, respectively.

Algorithm 2 presents an iterative BNNR scheme for solving the model (2). After performing BNNR algorithm, we can obtain a completed matrix $M^*$, where all the unknown entries of matrix $M$ have been filled out.

**Algorithm 2**: BNNR Algorithm

**Input:** The to-be-complete $M$, parameters $\alpha$, and $\beta$.
**Output:** Completed matrix $M^*$.
1. initialize $X_1 = P_\Omega(M)$, $W_1 = X_1$, $Y_1 = X_1$;
2. $k \leftarrow 1$;
**repeat**
3. $W_{k+1} \leftarrow \mathcal{Q}_{[0,1]}(W^*)$;
4. $X_{k+1} \leftarrow \mathcal{D}_{\frac{1}{\beta}}\left(W_{k+1} - \frac{1}{\beta}Y_k\right)$;
5. $Y_{k+1} \leftarrow Y_k + \beta(X_{k+1} - W_{k+1})$;
6. $k \leftarrow k + 1$;
**until convergence**
7. $M^* = W_k$;
8. **return** $M^*$.

**OMC2 algorithm.** We propose the OMC algorithm for bilayer networks (OMC2) to predict the potential drug–disease associations, whose goal is to obtain the low-rank matrices of drug–disease relationships from drug-side information and disease-side information. Firstly, we combine the updated disease–drug association matrix with the drug similarity matrix and create a block adjacency matrix $M_1$, as illustrated in Fig 1(d). Meanwhile, from the disease-side, we combine the updated disease–drug association matrix with the disease similarity matrix and generate a block adjacency matrix $M_2$, as illustrated in Fig 1(e). Secondly, the BNNR algorithm is implemented to fill out the unknown entries of $M_1$ and $M_2$. Finally, we calculate the average of two predicted drug–disease association matrices to obtain the final predicted matrix $A_{DR}^*$. Each element $(A_{DR}^*)_{ij}$ represents the predicted score between disease $d_i$ and drug $r_j$. The higher the score, the more likely that the association exists. To identify the promising candidate indicates for a specific drug, we rank all candidates according to their scores in descending order. The detail of the OMC2 algorithm is described in Algorithm 3.

**Algorithm 3**: OMC2 Algorithm

**Input:** The drug similarity matrix $A_{RR} \in \mathbb{R}^{m \times m}$, the disease similarity matrix $A_{DD} \in \mathbb{R}^{n \times n}$, the disease-drug association matrix $A_{DR} \in \mathbb{R}^{n \times m}$, parameters $K$, $\alpha$, and $\beta$.
**Outout:** Predicted association matrix $A_{DR}^*$.
1. $A_{DR1} \leftarrow KNN\ preprocessing(A_{DR}, A_{DD}, K)$;
2. $M_1 = \begin{bmatrix} A_{RR} \\ A_{DR1} \end{bmatrix}$;
3. $A_{DR2} \leftarrow KNN\ preprocessing(A_{DR}, A_{RR}, K)$;
4. $M_2 = [A_{DR2}\quad A_{DD}]$;
5. $\begin{bmatrix} A_{RR}^* \\ A_{DR1}^* \end{bmatrix} \leftarrow BNNR(M_1, \alpha, \beta)$;
6. $[A_{DR2}^*\quad A_{DD}^*] \leftarrow BNNR(M_2, \alpha, \beta)$;
7. $A_{DR}^* = \frac{A_{DR1}^* + A_{DR2}^*}{2}$;
8. **return** $A_{DR}^*$.

## OMC algorithm for tri-layer networks

OMC can be easily extended from bilayer networks (OMC2) to tri-layer networks (OMC3) algorithm, where the disease–protein and drug–protein association information are incorporated to further improve prediction accuracy. Firstly, we collect drug–protein (target) interactions and disease–protein associations from different databases. This step has been discussed in the previous section. Secondly, based on the two bilayer networks, i.e., the DrNet-Dis network and the DisNet-Dr network, we design two corresponding tri-layer networks. We integrate protein nodes and drug–protein associations into the DrNet-Dis network and construct a drug–protein–disease network called DrNet-Pro-Dis, as showed in Fig 2(e). The block adjacency matrix of this tri-layer network is defined as

$$M_1 = \begin{bmatrix} A_{RR} \\ A_{PR} \\ A_{DR1} \end{bmatrix}.$$

Similarly, we integrate protein nodes and disease–protein associations into the DisNet-Dr network and create another tri-layer network called DisNet-Pro-Dr, as illustrated in Fig 2(f). The block adjacency matrix of DisNet-Pro-Dr network is defined as

$$M_2 = \begin{bmatrix} A_{DR2} & A_{DP} & A_{DD} \end{bmatrix}.$$

Thirdly, the BNNR algorithm is carried out to fill out the missing entries of $M_1$ and $M_2$ to obtain two predicted drug–disease association matrices. Finally, we calculate the average of these two matrices as the final output. The detail of OMC3 the algorithm is described in Algorithm 4.

**Algorithm 4**: OMC3 Algorithm

**Input:** Drug similarity matrix $A_{RR} \in \mathbb{R}^{m \times m}$, disease similarity matrix $A_{DD} \in \mathbb{R}^{n \times n}$, protein–drug association matrix $A_{PR} \in \mathbb{R}^{s \times m}$, disease-protein association matrix $A_{DP} \in \mathbb{R}^{n \times t}$, disease-drug association matrix $A_{DR} \in \mathbb{R}^{n \times m}$, parameters $K$, $\alpha$, and $\beta$.

**Output:** Predicted association matrix $A_{DR}^*$.

1. $A_{DR1} \leftarrow KNN\ preprocessing(A_{DR}, A_{DD}, K)$;

2. $M_1 = \begin{bmatrix} A_{RR} \\ A_{PR} \\ A_{DR1} \end{bmatrix}$;

3. $A_{DR2} \leftarrow KNN\ preprocessing(A_{DR}, A_{RR}, K)$;

4. $M_2 = \begin{bmatrix} A_{DR2} & A_{DP} & A_{DD} \end{bmatrix}$;

5. $\begin{bmatrix} A_{RR}^* \\ A_{PR}^* \\ A_{DR1}^* \end{bmatrix} \leftarrow BNNR(M_1, \alpha, \beta)$;

6. $\begin{bmatrix} A_{DR2}^* & A_{DP}^* & A_{DD}^* \end{bmatrix} \leftarrow BNNR(M_2, \alpha, \beta)$;

7. $A_{DR}^* = \frac{A_{DR1}^* + A_{DR2}^*}{2}$;

8. **return** $A_{DR}^*$.

## Results

In this section, we systematically evaluate the performance of our proposed methods (OMC2 and OMC3) for predicting drug-associated indications. First of all, several evaluation metrics are introduced and parameter settings are discussed. In order to compare our methods with several state-of-the-art approaches, we perform 10-fold cross-validation and *de novo* tests in

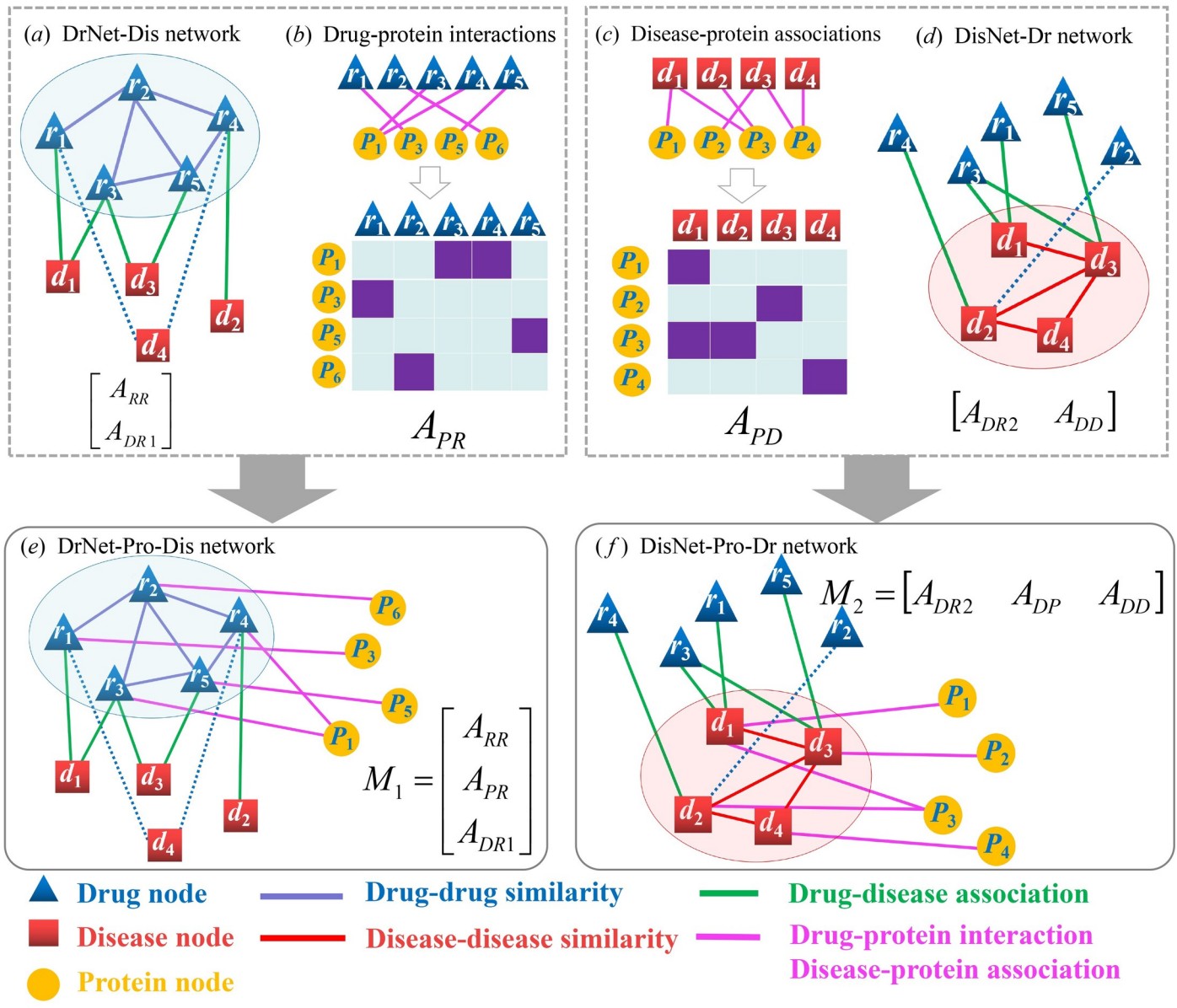

**Fig 2. The workflow of constructing the DrNet-Pro-Dis network and the DisNet-Pro-Dr network.** (a) DrNet-Dis network and its similarity matrix. (b) Drug–protein interactions and corresponding adjacency matrix. (c) Disease–protein associations and corresponding adjacency matrix. (d) DisNet-Dr network and its block adjacency matrix. (e) DrNet-Pro-Dis network and its block adjacency matrix. (f) DisNet-Pro-Dr network and its block adjacency matrix.

the gold standard dataset. Case studies are conducted to confirm the reliability of OMC3 in practical applications. Then, the performance of OMC and comparison on bilayer and tri-layer networks are discussed. Finally, we perform the same experiments on two other datasets to further illustrate the effectiveness and robustness of OMC2 and OMC3.

## Evaluation metrics

To evaluate the performance of our approaches, a 10-fold cross-validation experiment is conducted to identify candidate diseases for specific drugs. In the gold standard dataset, all approved drug–disease associations are randomly divided into ten parts with approximately

equal sizes. Each part is treated as the testing set in turn, and the training set is comprised of the remaining nine parts. To obtain convincing results, the 10-fold cross-validation is repeated 10 times and the final result is showed by the average value of the 10 folds. After the performing prediction, all candidate diseases associating with the test drug are ranked by their predicted scores in descending order. For a given rank threshold, the candidate disease is considered as a True Positive (TP) if its rank is above the threshold; otherwise, it is treated as a False Negative (FN). On the other hand, if the rank of a candidate disease had no association with the test drug is greater than the threshold, it is considered as a False Positive (FP), otherwise, it is treated as a True Negative (TN). Based on varying ranking thresholds, we can calculate True Positive Rate (TPR) and False Positive Rate (FPR) by

$$TPR = \frac{\#\ of\ TPs}{\#\ of\ TPs + \#\ of\ FNs}, \quad FPR = \frac{\#\ of\ FPs}{\#\ of\ FPs + \#\ of\ TNs},$$

and draw a Receiver Operating Characteristic (ROC) curve. Meanwhile, the area under the ROC curve (AUC) is utilized to evaluate the overall performance of a method. Precision and recall (equivalent to TPR) could be obtained to plot the precision-recall (PR) curve [25]. Due to the fact that the top-ranked result is a more important measurement in real-life drug-repositioning applications, the number of the retrieved correct associations is reported under different top ranking values.

## Parameter settings

In OMC2 and OMC3 algorithms, there are three hyper parameters to be determined, including $\alpha$, $\beta$, and $K$. In this subsection, using the OMC2 algorithm as an example, we explain the procedure of determining these parameters. The similar parameter determination procedure can be extended to the OMC3 algorithm.

For $\alpha$ and $\beta$, we perform a 10-fold cross-validation to find the most appropriate values by the grid search, which are chosen from {0.1, 1, 10, 100}. When the neighborhood size $K$ is fixed to 1, S1 Table shows the AUC values of OMC2 under different values of $\alpha$ and $\beta$ on the gold standard dataset. Our results show that the best performance is achieved by $\alpha = 1$ and $\beta = 10$.

For $K$, we firstly assign 1 and 10 to $\alpha$ and $\beta$, respectively and then use cross validation to pick an appropriate $K$ value from {1, 5, 10, 15, 20, 25, 30}. S1 Fig shows the AUC values of OMC2 under this setting. When $K$ is 10, the best AUC value is achieved. Since the values of $K$ have little effect on AUC values, we can treat $K = 10$ as a prior knowledge in other datasets for simplicity. Actually, We fixed the neighborhood size K to 10, the optimal values of $\alpha$ and $\beta$ are also equal to 1 and 10, respectively. The results are shown in S2 Table and it could further illustrate the stability of the parameter values.

Based on the above analysis, we finally choose $\alpha = 1$, $\beta = 10$, and $K = 10$ for the gold standard dataset as the default parameters.

## Comparison with other methods

In order to obtain convincing and fair comparison results, OMC2 and OMC3 are compared with the five state-of-the-art approaches: BNNR [14], DRRS [12], MBiRW [6], DrugNet [5], and HGBI [3]. The parameters in the compared approaches are set to either the default values in their papers or the best value by the grid search, if the default values are not provided. We rank the predicted indications and plot the ROC curves and PR curves to analyze the 10-fold cross-validation results.

As shown in Fig 3, OMC2 and OMC3 outperform the other methods in ROC curves, PR curves, and top-ranked results. More specifically, OMC2 and OMC3 obtain AUC values of

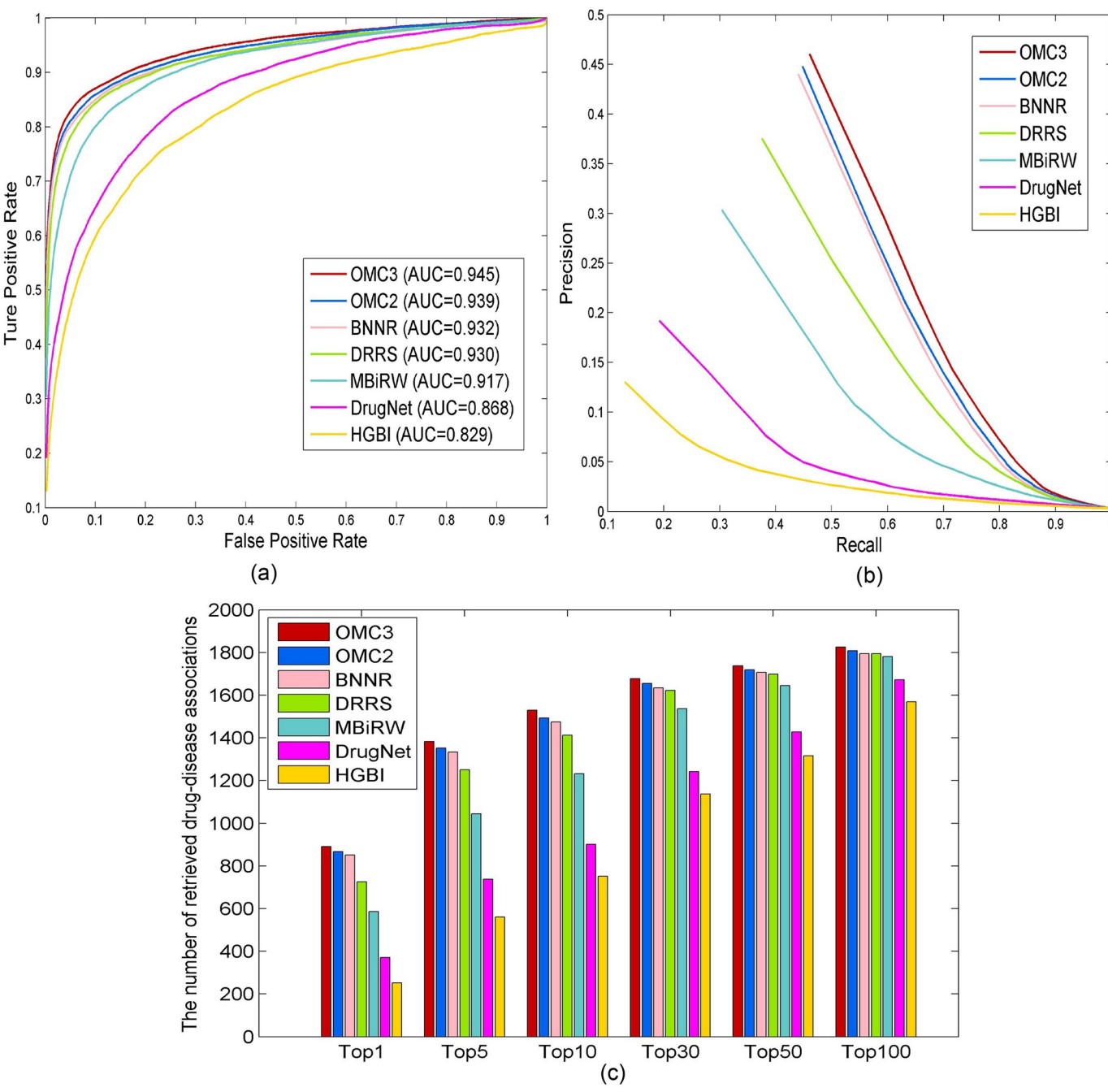

**Fig 3. The performance of all methods for predicting drug–disease associations in the 10-fold cross-validation.** (a) ROC curves of prediction results. (b) PR curves of predicting candidate diseases for drugs. (c) The number of correctly retrieved drug–disease associations for various rank thresholds.

0.939 and 0.945, while BNNR, DRRS, MBiRW, DrugNet, and HGBI yield AUC values of 0.932, 0.930, 0.917, 0.868, and 0.829, respectively. In real-life drug-repositioning applications, researchers particularly care about precision, because the precise prediction can significantly reduce experimental cost and time. The PR curves show OMC2 and OMC3 achieve the second best and the best precisions of 0.449 and 0.461, while BNNR, DRRS, MBiRW, DrugNet, and HGBI have the precisions of 0.440, 0.375, 0.304, 0.192, and 0.130, respectively. It is important

to note that OMC3 can successfully prioritize 46.1% true drug–disease associations at top rank. A true drug–disease association is treated as the retrieved correct association when its predicted rank is higher than the specified ranking threshold. The numbers of correct associations predicted by all methods under different top ranking values are shown in Fig 3(c). The numbers of retrieved associations of both OMC2 and OMC3 exceed those of the other competing approaches. Specifically, among 1, 933 true drug–disease associations, 1, 493(77.2%) and 1, 529(79.1%) associations are correctly predicted at top 10 by OMC2 and OMC3, while in comparison, 1, 475(76.3%), 1, 413(73.1%), 1, 232(63.7%), 900(46.6%), and 752(38.9%) associations are identified by BNNR, DRRS, MBiRW, DrugNet, and HGBI, respectively.

## Prediction of potential indications for new drugs

To evaluate the performance of OMC2 and OMC3 for identifying indications of novel drugs, we conduct a *de novo* test, where these drugs with only one known drug–disease association are chosen. For each of these drugs, the unique disease association is removed in turn as the test sample, and other known drug–disease associations are used as the training samples. There are totally 171 drugs with only one known associated disease in gold standard dataset.

As shown in Fig 4, OMC2 and OMC3 achieve the AUC values of 0.851 and 0.871, while BNNR, DRRS, MBiRW, DrugNet, and HGBI have inferior results with the AUC values of 0.830, 0.824, 0.818, 0.782, and 0.746, respectively. OMC3 has demonstrated its advantages measured by PR curves. For top-ranked results, OMC3 outperforms all methods at all ranking thresholds. Meanwhile, OMC2 surpasses the compared approaches at top 5, 10, 30, 50 and 100, except for being inferior to DRRS at top 1. Specifically, 74 and 88 drugs are identified correctly at top 5 by OMC2 and OMC3, respectively. In comparison, 73, 62, 71, 52, and 36 drugs are predicted by BNNR, DRRS, MBiRW, DrugNet, and HGBI, respectively. Summarizing the above results, one can find that our OMC methods are effective to address the cold-start problem to identify potential indications for novel drugs. In particular, OMC3 yields further improvement over OMC2, indicating the effectiveness of incorporating target association information in the tri-layer network.

## Case studies

We apply OMC3 to predict new uses for already approved drugs in real applications. To predict novel indications for existing drugs in the gold standard dataset, we consider all known associations between drugs and diseases as the training samples and the unknown drug–disease pairs as the candidate samples. By carrying out the OMC3 algorithm, the predicted scores of all candidate pairs are obtained and sorted for each specific drug.

In order to verify the predicted diseases, we choose three representative drugs: Doxorubicin, Flecainide, and Levodopafour. We confirm the potential diseases associated with the given drug by retrieving authoritative public databases, such as CTD [21], DrugBank, and KEGG [26]. The newly predicted indications and their supporting evidences are listed in Table 1. One can find that more than three novel indications are validated on top-5 for each representative drug. As shown in this case study, OMC3 can be used as an effective method for identifying new indications for specific drugs in practical applications. In order to provide more helpful references for medical researchers, the top-30 candidate indications of each drug are listed in S3 Table.

## Effectiveness of OMC on performance

In order to evaluate the effectiveness of OMC, we compare OMC2 with algorithms using only drug- or disease-side information in 10-fold cross-validation. The first algorithm, called

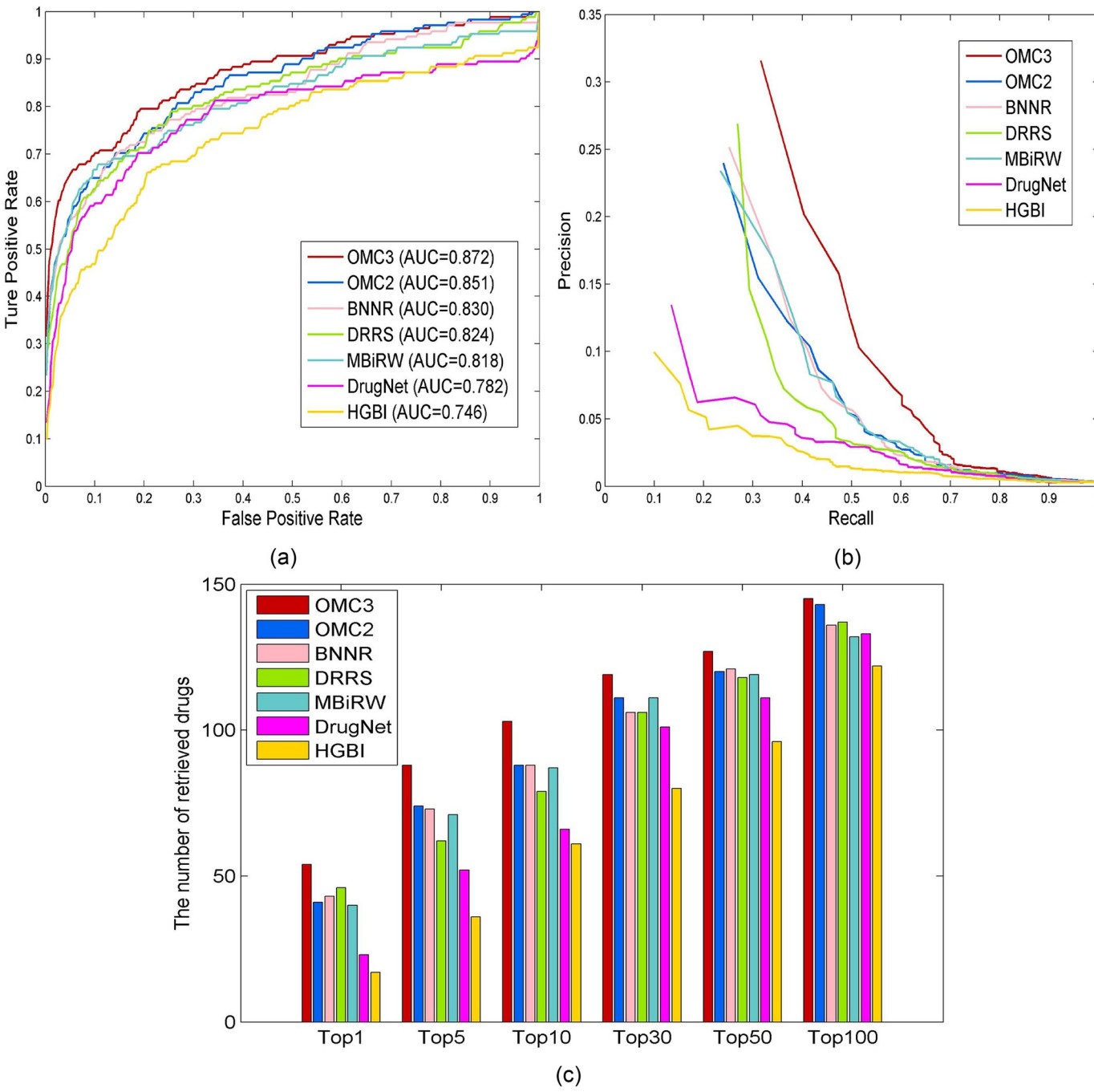

**Fig 4. Performance of all methods in predicting potential diseases for new drugs.** (a) ROC curves of prediction results. (b) PR curves of predicting candidate diseases for drugs. (c) Number of correctly retrieved drug–disease associations for various rank thresholds.

OMC-drug, is to obtain $A_{DR1}$ by BNNR in DrNet-Dis network, while the other one, called OMC-disease, is to recover $A_{DR2}$ by BNNR in DisNet-Dr network. As shown in S2 Fig, both OMC-drug and OMC-disease are inferior to OMC2 in each fold in terms of AUC. In conclusion, consolidating drug- and disease-side associations in OMC2 is a better way to predict drug–disease associations than just using one-side information.

**Table 1. The top-5 candidate diseases for Doxorubicin, Flecainide, and Levodopa.**

| Drugs (DrugBank IDs) | Top-5 candidate diseases (OMIM IDs) | Evidences |
|---|---|---|
| Doxorubicin (DB00997) | Dohle bodies (223350) | |
| | Reticulum cell sarcoma (267730) | CTD |
| | Small cell cancer of the lung (182280) | CTD |
| | Leukemia (109543) | CTD/DB/KEGG |
| | Testicular germ cell tumor (273300) | CTD |
| Flecainide (DB01195) | Atrial fibrillation (608583) | CTD |
| | Cardiac arrhythmia (115000) | CTD/DB |
| | Diastolic hypertension (608622) | CTD |
| | Hyperplastic myelinopathy (147530) | |
| | Nephropathy-hypertension (161900) | |
| Levodopa (DB01235) | Parkinson disease (168600) | CTD/DB/KEGG |
| | Dementia (125320) | CTD/DB |
| | Schizophrenia (181500) | CTD |
| | Optic atrophy, hearing loss, and peripheral neuropathy (165199) | CTD |
| | Hyperplastic myelinopathy (147530) | |

## Comparison on tri-layer networks

In this subsection, we illustrate the performance and the computational efficiency of different approaches in tri-layer networks. BNNR, DRRS, DrugNet, and HGBI algorithms are taken into account for extending from bilayer networks into tri-layer networks, in comparison with OMC3. Since the protein association information is incorporated, the resulted affinity matrix of the tri-layer network is significantly enlarged. This also poses computational challenges in the factorization algorithms in matrix completion, which often grow cubically. The running time of each approach is obtained on a Linux server with CPU 2.30 GHz and 128 GB memory.

As described in our previous works, BNNR and DRRS constructed the same bilayer networks between drugs and diseases. In order to construct a tri-layer heterogeneous network, we integrate protein-related information into the network, including protein–protein similarities, drug–protein interactions, and disease–protein associations. Accordingly, we get the corresponding square, symmetric adjacency matrix defined as follows,

$$M = \begin{bmatrix} A_{RR} & A_{DR}^T & A_{PR}^T \\ A_{DR} & A_{DD} & A_{PD}^T \\ A_{PR} & A_{PD} & A_{PP} \end{bmatrix},$$

where $A_{PP}$ represents the protein–protein similarity matrix, which is calculated based on the amino acid sequence alignment by Rcpi [27]. The programs for completing the matrix $M$ by BNNR and DRRS are called BNNR3 and DRRS3, respectively. For DrugNet, it is also applied to tri-layer networks by integrating target-related information [5], which is denoted as DrugNet3 here. DrugNet3 can predict drug–disease relationships by propagating information in the drug–target–disease network. Based on the guilt-by-association principle, the authors of HGBI had extended bilayer networks into tri-layer networks by integrating drug, target, and disease information [4], which was called TL-HGBI (denotes HGBI3 here).

The 10-fold cross-validation is uniformly conducted in the same gold standard dataset for OMC3, BNNR3, DRRS3, DrugNet3, and HGBI3. As shown in Fig 5(a) and 5(b), OMC3

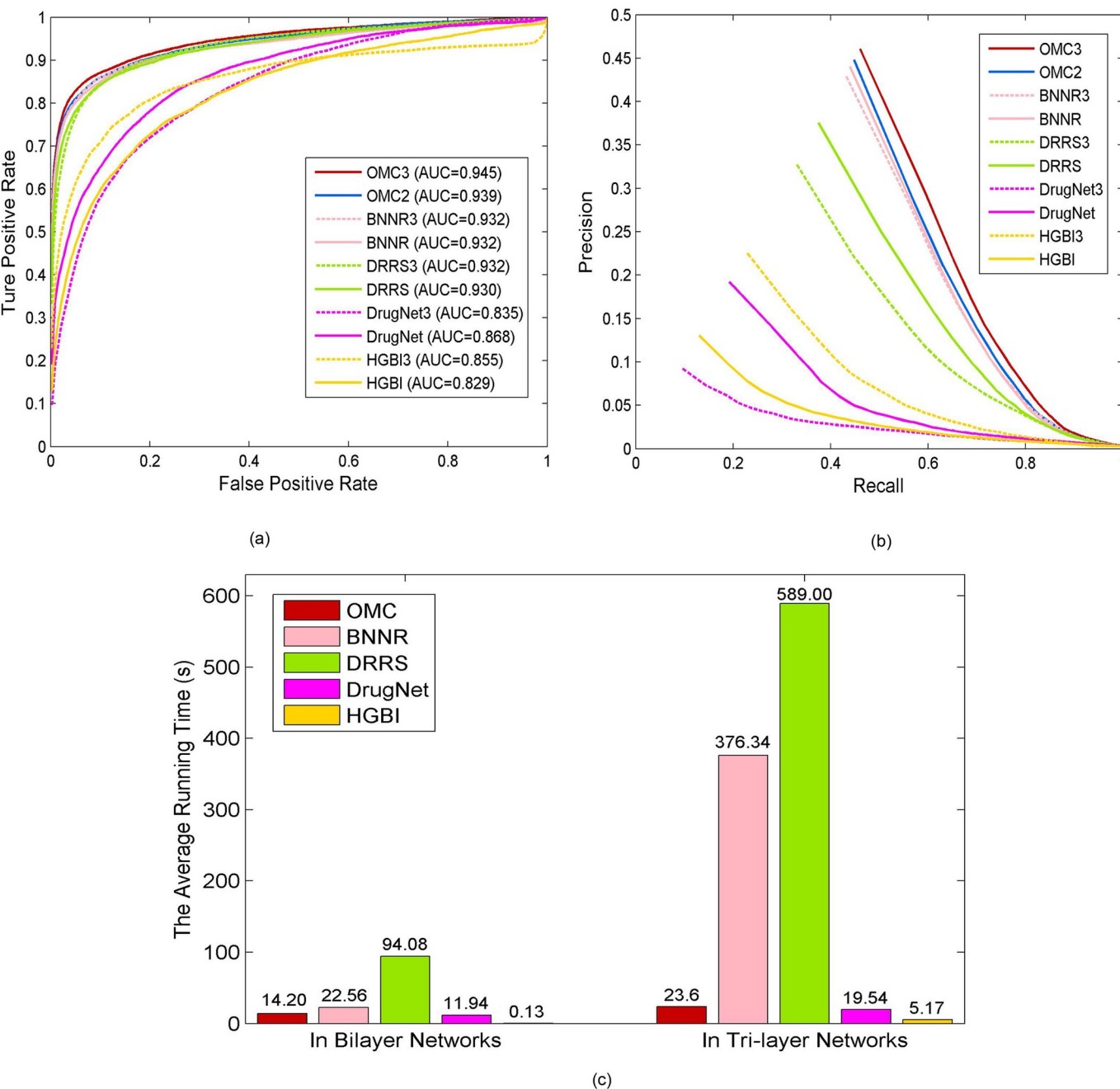

**Fig 5. Method comparison in bilayer networks and tri-layer networks.** (a) ROC curves of prediction results. (b) PR curves of prediction results. (c) The average running time of each fold in the 10-fold cross-validation.

outperforms the other approaches measured by the AUC values of the ROC curves and the precision. Specifically, OMC3 obtains the best AUC value of 0.945, while BNNR3, DRRS3, DrugNet3, and HGBI3 have the AUC values of 0.932, 0.932, 0.835, and 0.855, respectively. The PR curves show that OMC3 obtains the best precision with 0.460, while BNNR3, DRRS3, DrugNet3, and HGBI3 have the precision values of 0.431, 0.329, 0.093, and 0.227, respectively.

Surprisingly, a method extended from bilayer networks into tri-layer networks does not necessary improve the prediction performance. In fact, only OMC and HGBI obtain performance improvement in tri-layer networks over bilayer ones. BNNR3, DRRS3, and DrugNet3 yield even worse performance when tri-layer networks are used compared to the corespondent algorithms on bilayer networks. This is due to the fact that protein–protein similarities calculated by the algorithm contain a large amount of "noise", which causes BNNR3, DRRS3, and DrugNet3 to degrade their prediction performance. In contrast, OMC3 avoids the use of protein–protein similarities and the information OMC3 used is experimentally proven, such as drug–protein interactions and disease–protein associations, which in turn leads to performance improvement over OMC2 on bilayer networks.

As shown in Fig 5(c), the average running time of BNNR and DRRS increase sharply from bilayer networks (BNNR2 and DRRS2) to tri-layer networks (BNNR3 and DRRS3), due to the increase of the affinity matrix. Nevertheless, this does not have such a significant impact on OMC, DrugNet, and HGBI. For OMC, this is because OMC keeps the matrix completion computation at the bilayer network level. As a result, OMC is not only better in terms of the prediction performance, but is also computational efficiency.

## Experiments on the other datasets

We apply OMC2 and OMC3 to two other datasets, including Cdataset [6] and DNdataset [5], to demonstrate their robustness. Cdataset contains 663 drugs collected in DrugBank, 409 diseases obtained in OMIM database, and 2, 352 known drug–disease associations. In addition, we have collected drug–protein interactions related to drugs of Cdataset from DrugBank and retrieved a total of 3, 251 associations between 637 drugs and 891 proteins. For disease–protein associations, we download disease–gene interactions related to diseases of Cdataset from CTD database, and map genes into proteins in the UniprotKB database. There are 1, 280 associations between 226 diseases and 1, 002 proteins. The drug similarity and disease similarity are calculated in the same way as described in the previous section. DNdataset includes 1, 490 drugs registered in DrugBank, 4, 516 diseases annotated by Disease Ontology (DO) terms, 18, 107 proteins extracted from BioGRID, 11, 658 disease–protein associations directly extracted from the disease and gene annotations (DGA), 4, 026 drug–protein interactions collected in DrugBank, and 1, 008 known drug–disease associations. We evaluate the performance of our methods on Cdataset and DNdataset by performing a 10-fold cross-validation and *de novo* experiments.

For Cdataset, as shown in S3(a)–S3(c) Fig, OMC2 and OMC3 demonstrate superior performance in terms of ROC curve, PR curve, and top-ranked results in the 10-fold cross-validation. Specifically, OMC2 and OMC3 obtain the AUC values of 0.953 and 0.957 in the ROC curves, while BNNR, DRRS, MBiRW, DrugNet, and HGBI have 0.948, 0.947, 0.933, 0.903, and 0.858, respectively. The PR curves indicate that OMC2 and OMC3 achieve the second best precision of 0.476 and the best precision of 0.489, while the precision values in BNNR, DRRS, MBiRW, DrugNet, and HGBI are 0.471, 0.403, 0.351, 0.239, and 0.168, respectively. In addition, OMC2 and OMC3 outperform the other methods in the top-ranked results with respect to different ranking thresholds. In the *de novo* test, there are 177 drugs with only one known associated disease in Cdataset. As shown in S4(a)–S4(c) Fig, OMC2 and OMC3 obtain the AUC values of 0.830 and 0.846, respectively, while BNNR, DRRS, MBiRW, DrugNet, and HGBI have the AUC values of 0.812, 0.819, 0.804, 0.785, and 0.732, respectively. Both OMC2 and OMC3 exceed the other methods in terms of AUC values as well. For top-ranked results, among 177 test drugs, 100 (56.5%) drugs are correctly identified at top 10 by OMC3, while

only 87 (49.2%), 78 (44.1%), 80 (45.2%), 61 (34.5%), and 48 (27.1%) drugs are predicted by BNNR, DRRS, MBiRW, DrugNet, and HGBI, respectively.

For DNdataset, in the 10-fold cross-validation results shown in S5(a)–S5(c) Fig, OMC2 and OMC3 obtain the AUC values of 0.957 and 0.965, while BNNR, DRRS, MBiRW, DrugNet, and HGBI yield the AUC values of 0.955, 0.934, 0.956, 0.950, and 0.921, respectively. Similar to that of Cdataset, OMC2 obtains the second best precision of 0.360 and OMC3 obtains the best precision of 0.369 in PR curves. Moreover, OMC2 and OMC3 outperform the other methods on top-ranked results at different ranking thresholds. In the *de novo* test, OMC3 also outperforms the other methods. As shown in S6(a)–S6(c) Fig, OMC2 and OMC3 obtain the AUC values of 0.963 and 0.972, while BNNR, DRRS, MBiRW, DrugNet, and HGBI have the AUC values of 0.956, 0.946, 0.970, 0.969, and 0.928, respectively. For top-ranked results, among 347 test drugs, 228 (65.7%) and 231 (66.6%) drugs are correctly identified at top 1 by OMC2 and OMC3, while only 218 (62.8%), 213 (61.4%), 219 (63.1%), 156 (45.0%), and 150 (43.2%) drugs are predicted by BNNR, DRRS, MBiRW, DrugNet, and HGBI, respectively. In summary, the above results on Cdataset and DNdataset demonstrate the robustness and generalization of OMC.

## Discussion

In this study, we have proposed a novel OMC method for predicting drug-associated indications, which can effectively integrate multiple types of drug and disease information. In addition, our method can be simply extended from bilayer networks to tri-layer networks by incorporating drug-target associations. Furthermore, OMC effectively avoids the use of noisy data in tri-layer networks. The performance of our methods (OMC2 and OMC3) are validated by the cross validation, *de novo* experiments, and case studies. The experimental results indicate that our methods are effective compared with the latest approaches, particularly for *de novo* drugs.

However, OMC has two potential limitations. First, the drug and disease similarity computations in this work may be not optimal. More reliable similarity measures, for example consensus integrating multiple similarities computations from different aspects could improve the performance of OMC. Second, OMC must perform matrix completion twice from both drug-side and disease-side before the final predicted score is obtained.

OMC can actually be used on other drug-related predictions, such as synergistic drug combination and small molecule–miRNA association prediction. The synergistic drug combination is based on the assumption that principal drugs which obtain the synergistic effect with similar adjuvant drugs are often similar and vice versa [28]. That means the drug combination matrix is also of low-rank. Therefore, OMC can be applied to predict potential synergistic drug combinations by integrating the drug similarity matrix and the drug–target interaction matrix. In addition, it may avoid classifying principal drugs and adjunct drugs before obtaining the final score of drug combinations. MiRNAs play an important role in the initiation and development of various human diseases. Several drug-like compound libraries targeting different miRNAs have been successfully screened in cell assays, further demonstrating the possibility of targeting miRNAs with small molecules. Hence, it is very meaningful and promising to develop computational models for drug repositioning based on drug related miRNA. Some original and novel methods have been proposed in recent years [29]. Especially, based on tri-layer heterogeneous networks, more prior information is used to obtain better prediction performance [30]. In the future, we plan to extend our OMC method to explore drug combinations and miRNA-small molecule associations for drug repositioning.

## Supporting information

**S1 Fig. The AUC values are indicated by the OMC2 algorithm when the neighborhood size** $K$ **is chosen from {1, 5, 10, 15, 20, 25, 30} in the cross validation.**
(TIF)

**S2 Fig. Performance comparison of the OMC2, OMC-drug and OMC-disease in the 10-fold cross-validation in terms of AUC.** The result of each fold is presented.
(TIF)

**S3 Fig. (a) ROC curves of prediction results. (b) PR curves of predicting candidate diseases for drugs. (c) The number of correctly retrieved drug–disease associations for various rank thresholds.** The performance of all methods for predicting drug–disease associations in the 10-fold cross-validation on CDataset.
(TIF)

**S4 Fig. (a) ROC curves of prediction results. (b) PR curves of predicting candidate diseases for drugs. (c) The number of correctly retrieved drug–disease associations for various rank threshold.** The performance of all methods in predicting potential diseases for new drugs on CDataset.
(TIF)

**S5 Fig. (a) ROC curves of prediction results. (b) PR curves of predicting candidate diseases for drugs. (c) The number of correctly retrieved drug–disease associations for various rank thresholds.** The performance of all methods for predicting drug–disease associations in the 10-fold cross-validation on DNdataset.
(TIF)

**S6 Fig. (a) ROC curves of prediction results. (b) PR curves of predicting candidate diseases for drugs. (c) The number of correctly retrieved drug–disease associations for various rank threshold.** The performance of all methods in predicting potential diseases for new drugs on DNdataset.
(TIF)

**S1 Table. The AUC values under different values of** $\alpha$ **and** $\beta$ **in the 10-fold cross-validation for the gold standard dataset.**
(DOC)

**S2 Table. The AUC values based on K = 10 for** $\alpha$ **and** $\beta$ **in the 10-fold cross-validation for the gold standard dataset.**
(DOC)

**S3 Table. The top-30 candidate indications of all drugs listed by OMC3.**
(XLSX)

## Author Contributions

**Data curation:** Mengyun Yang, Huimin Luo.

**Funding acquisition:** Mengyun Yang, Huimin Luo, Jianxin Wang.

**Investigation:** Mengyun Yang, Huimin Luo.

**Methodology:** Mengyun Yang.

**Resources:** Mengyun Yang, Huimin Luo.

**Software:** Mengyun Yang, Huimin Luo.

**Supervision:** Jianxin Wang.

**Validation:** Mengyun Yang, Huimin Luo.

**Writing – original draft:** Mengyun Yang.

**Writing – review & editing:** Yaohang Li, Fang-Xiang Wu, Jianxin Wang.

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
