## [Decision Letter · Decision Letter 0]

4 Sep 2019

Dear Dr Wang,

Thank you very much for submitting your manuscript 'Overlap matrix completion for predicting drug-associated indications' for review by PLOS Computational Biology. Your manuscript has been fully evaluated by the PLOS Computational Biology editorial team and in this case also by independent peer reviewers. The reviewers appreciated the attention to an important problem, but raised some substantial concerns about the manuscript as it currently stands. While your manuscript cannot be accepted in its present form, we are willing to consider a revised version in which the issues raised by the reviewers have been adequately addressed. We cannot, of course, promise publication at that time.

Sincerely,

Edwin Wang

Benchmarking Editor

PLOS Computational Biology

Nir Ben-Tal

Deputy Editor

PLOS Computational Biology

[LINK]

Reviewer's Responses to Questions

**Comments to the Authors:**

Reviewer #1: This paper formulates computational drug repositioning problem based on matrix completion. An overlap matrix completion method is proposed to predict the potential drug-associated indications by integrating reliable drug-related and disease-related information in bilayer and tri-layer networks. Experimental studies on three different datasets show that the proposed approach can outperform several baseline methods, especially in tri-layer networks.

1. When I take a look at case studies, only three drugs are used as examples. In order to give more references for medical researchers, the predicted top-30 indications of each drug need to be listed in the supplementary file.

2. Could MBiRW algorithm be used for comparison in tri-layer networks? Please clarify it.

3. The first letter of 'hyperplastic' should be capitalized in Table 2.

4. There are some writing and grammatical errors in the manuscript. The author should examine and revise the article carefully.

5. The quality of pictures in the manuscript need to be improved.

6．Drug combination can be regarded as a new way of drug repositioning. Could you give some discussions whether your method could be used for synergistic drug combination prediction in the future (PMID: 27415801 should be discussed)? In addition, could you consider the possibility of developing computational models for drug repositioning based on drug related miRNA, a new kind of drug target? Some important studies should be discussed (PMIDs: 30325405 and 29943160).

7. Please analyze the novelty and limitations of the method in the section of Conclusions.

Reviewer #2: This manuscript proposed a new method, namely the overlap matrix completion (OMP) method, to find potential drug-associated relationship. Numerical results show that the new algorithm can achieve accurate results with high efficiency. This is a well-written and well-organized work to address an important issue in computational biology.

Major issue:

The computation to find the optimal values of alpha, beta and K may not be optimal. First the samples are not evenly distributed, for example, alpha \\in [0.1. 1 10, 100]. It may be possible the optimal value is actually around 2 or around 5. In addition, the work first fixed K=1 and found the optimal value of alpha=1, beta=10. Then based on alpha=1, beta=10, found the optimal value was K=10. The question is, if K=10, what is the optimal values of alpha and beta? Is there any possibility to develop an iterated scheme to search for the optimal values?

Minor issue

There are many curves in each figure, and 11 figures in this work. It may be better to put some figures in supplementary information and show the remaining figures in larger size.

**Have all data underlying the figures and results presented in the manuscript been provided?**

Reviewer #1: Yes

Reviewer #2: Yes

PLOS authors have the option to publish the peer review history of their article (what does this mean?). If published, this will include your full peer review and any attached files.

Reviewer #1: No

Reviewer #2: No

---

## [Decision Letter · Decision Letter 1]

12 Nov 2019

Dear Dr Wang,

We are pleased to inform you that your manuscript 'Overlap matrix completion for predicting drug-associated indications' has been provisionally accepted for publication in PLOS Computational Biology.

In the meantime, please log into Editorial Manager at https://www.editorialmanager.com/pcompbiol/, click the "Update My Information" link at the top of the page, and update your user information to ensure an efficient production and billing process.

One of the goals of PLOS is to make science accessible to educators and the public. PLOS staff issue occasional press releases and make early versions of PLOS Computational Biology articles available to science writers and journalists. PLOS staff also collaborate with Communication and Public Information Offices and would be happy to work with the relevant people at your institution or funding agency. If your institution or funding agency is interested in promoting your findings, please ask them to coordinate their releases with PLOS (contact ploscompbiol@plos.org).

Thank you again for supporting Open Access publishing. We look forward to publishing your paper in PLOS Computational Biology.

Sincerely,

Edwin Wang

Benchmarking Editor

PLOS Computational Biology

Nir Ben-Tal

Deputy Editor

PLOS Computational Biology

<br \\>

Reviewer's Responses to Questions

**Comments to the Authors: **

Reviewer #1: satisfactory revisions have been implemented

Reviewer #2: Authors have addressed the issues in the first report very well. The quality of this manuscript has improved much. There is no further comment.

**Have all data underlying the figures and results presented in the manuscript been provided?**

Reviewer #1: Yes

Reviewer #2: Yes

PLOS authors have the option to publish the peer review history of their article (what does this mean?). If published, this will include your full peer review and any attached files.

<br \\>

<br \\>

Reviewer #1: No

Reviewer #2: No

---

## [Editor Report · Acceptance letter]

11 Dec 2019

PCOMPBIOL-D-19-01173R1 

Overlap matrix completion for predicting drug-associated indications

Dear Dr Wang,

I am pleased to inform you that your manuscript has been formally accepted for publication in PLOS Computational Biology. Your manuscript is now with our production department and you will be notified of the publication date in due course.

With kind regards,

Sarah Hammond
